# *Limonium tetragonum* Promotes Running Endurance in Mice through Mitochondrial Biogenesis and Oxidative Fiber Formation

**DOI:** 10.3390/nu14193904

**Published:** 2022-09-21

**Authors:** Yong Gyun Lee, Mi-Young Song, Hwangeui Cho, Jong Sik Jin, Byung-Hyun Park, Eun Ju Bae

**Affiliations:** 1School of Pharmacy, Jeonbuk National University, Jeonju 54896, Jeonbuk, Korea; 2Department of Biochemistry and Research Institute for Endocrine Sciences, Jeonbuk National University Medical School, Jeonju 54896, Jeonbuk, Korea; 3Department of Oriental Medicine Resources, Jeonbuk National University, Iksan 54596, Jeonbuk, Korea

**Keywords:** *Limonium tetragonum* water extract, smart-farming system, endurance exercise, mitochondrial biogenesis, slow myofiber formation, exercise mimetic

## Abstract

The purpose of this study was to examine whether *Limonium tetragonum*, cultivated in a smart-farming system with LED lamps, could increase exercise capacity in mice. C57BL/6 male mice were orally administered vehicle or *Limonium tetragonum* water extract (LTE), either 30 or 100 mg/kg, and were subjected to moderate intensity treadmill exercise for 4 weeks. Running distance markedly increased in the LTE group (100 mg/kg) by 80 ± 4% compared to the vehicle group, which was accompanied by a higher proportion of oxidative fibers (6 ± 6% vs. 10 ± 4%). Mitochondrial DNA content and gene expressions related to mitochondrial biogenesis were significantly increased in LTE-supplemented gastrocnemius muscles. At the molecular level, the expression of PGC-1α, a master regulator of fast-to-slow fiber-type transition, was increased downstream of the PKA/CREB signaling pathway. LTE induction of the PKA/CREB signaling pathway was also observed in C2C12 cells, which was effectively suppressed by PKA inhibitors H89 and Rp-cAMP. Altogether, these findings indicate that LTE treatment enhanced endurance exercise capacity via an improvement in mitochondrial biosynthesis and the increases in the formation of oxidative slow-twitch fibers. Future study is warranted to validate the exercise-enhancing effect of LTE in the human.

## 1. Introduction

Due to sedentary behavior and dietary changes, the risk for the development of overweight/obesity and the associated metabolic diseases is continuously increasing. Therefore, aerobic exercise as a way of lifestyle intervention has emerged as an effective means to prevent metabolic problems [1]. Exercise also decreases the risk of cardiovascular [2] and neurocognitive disease [3], as well as the risk of contracting some types of cancer [4], while increasing bone mineral density [5], quality of life [6], and even lifespan [7]. Adult skeletal muscle is composed of two types of myofibers, type I and type II, that possess different metabolic and contractile properties. Type I myofibers exhibit a high level of myosin heavy chain (MyHC) isoform I, robust oxidative capacity, abundant mitochondrial content, and are resistant to fatigue [8]. In contrast, type II fibers (MyHC IIb) have less mitochondria and oxidative capacity than type I fibers, and demonstrate fast-twitch contraction [9]. Metabolic and contractile properties of MyHC IIa and MyHC IIx are in-between those of MyHC I and MyHC IIb [8]. Mature skeletal muscle is highly plastic, as its fibers can adjust their MyHC isoform expression and mitochondrial content in response to exercise, electrical stimulation, disease, and other factors [10]. For example, resistance exercise promotes protein synthesis resulting in hypertrophy of fast-twitch fibers, whereas endurance exercise triggers mitochondrial biogenesis and affects the expression of muscle-fiber-specific proteins, thereby promoting fast-to-slow fiber-type switching [11]. Conversely, aging, a sedentary lifestyle, obesity, and type 2 diabetes decrease oxidative capacity of skeletal muscles [12]. In humans, endurance in daily life normally refers to the physical strength of a person to do certain things for a prolonged time, but it also means the ability to resist fatigue. Therefore, an improvement in endurance is proportional to fatigue resistance, and fast-to-slow myofiber-type transition can improve both endurance capacity and resistance to fatigue.

While the signaling pathways that govern muscle fiber-type conversion have yet to be fully elucidated, a number of studies have identified peroxisome proliferator-activated receptor (PPAR) gamma coactivator 1-alpha (PGC-1α) as a pivotal regulator of fast-to-slow fiber-type switching [13]. PGC-1α transcription is regulated by the binding of several transcription factors such as activating transcription factor 2 (ATF2) [14], forkhead Box O1 (FoxO1) [15], cyclic AMP response element-binding protein (CREB) [16], and myocyte enhancer factor 2 (MEF2) [17]. Upon cold exposure, cAMP-activated protein kinase A (PKA) phosphorylates p38 mitogen-activated protein kinase (p38 MAPK), which further phosphorylates and promotes the binding of ATF2 to the promoter of PGC-1α [14]. In turn, PGC-1α binds to and regulates several transcription factors, including PPARγ [16], PPARα [18], nuclear respiratory factor 1 [19], estrogen-related receptor alpha and gamma [20], and forkhead Box O1 [21], and enhances the expression of a series of genes related to mitochondrial biogenesis and oxidative phosphorylation. The transcriptional activity of PGC-1α is governed by the coordinated sequential actions of AMP-activated protein kinase (AMPK) and Sirt1. Upon exercise, AMPK phosphorylates PGC-1α at Thr177 and Ser538 [22], which primes it for subsequent deacetylation by Sirt1 at thirteen lysine residues [23]. 

*Limonium tetragonum* is a salt-tolerant biennial halophyte of the Plumbaginaceae family that grows widely in salt marshes and on muddy seashores along the southwestern coastal areas of South Korea. *L. tetragonum* possesses anti-oxidative properties and has been reported to have multiple beneficial effects against various pathologies such as high-fat diet-induced obesity [24] and alcohol-induced liver damage in mice [25], and diethylnitrosamine-induced liver fibrosis in rats [26]. In vitro studies have demonstrated that *L. tetragonum* extract suppresses the melanogenesis of B16-F10 melanoma cells [27] and the proliferation of HSC-T6 hepatic stellate cells, and inhibits matrix metalloproteinase activity in HT1080 fibrosarcoma cells [28]. Several bioactive flavonoids, flavonoid glycosides, and catechins which may produce pharmacological effects have been isolated from the ethyl acetate soluble fraction of *L. tetragonum* extract [29]. Due to *L. tetragonum*’s constituents and bioactivity, we further hypothesized that *L. tetragonum* may be able to increase exercise performance, which has yet to be explored. In this study, we prepared the water extract of *L. tetragonum* (LTE) using plants that had been cultivated in a smart-farm factory using LED grow lights, as opposed to those that grew naturally along the coast of South Korea.

We administered LTE to C57BL/6 mice along with chronic exercise training program, and then assessed the effect of combining exercise and LTE treatment on exercise endurance and mitochondrial oxidative capacity relative to exercise alone.

## 2. Materials and Methods

### 2.1. Cultivation of L. tetragonum in a Smart-Farming System

*L. tetragonum* seeds were identified and collected by Suk-Kyu Kim (Halopharm Co., Iksan, Korea) in the coastal area of Muan-gun, Korea. Seeds were cultivated for 40 days in a plant factory at the LED Agri-bio Fusion Technology Research Center (Jeonbuk National University, Iksan, Korea). The growth room temperature of 21 ± 1 °C was controlled by air conditioning and circulation fans. LED light conditions (red:blue = 6:4 ratio) were set to 150 μmol/m^2^/s at 20 cm with a 16 h (light)/8 h (dark) cycle. Relative humidity was maintained at 60 ± 5% during the cultivation period. Electrical conductivity and pH were kept at 2.2 ± 0.2 ms·cm^−1^ and 6.0 ± 0.5, respectively. All of these conditions were monitored by an environmental control system.

### 2.2. Preparation of LTE

The leaf of *L. tetragonum* was washed and then dried overnight in an oven at 60 °C. The dried leaf of *L. tetragonum* was extracted for 1 h in hot water (100 °C) according to a solid to liquid ratio of 1:25 (*w*/*v*) using a reflux condenser. The extract was filtered with Whatman filter paper No.1, lyophilized (batch method), and stored at 4 °C before use. The percentage yield of the dried extract was 28.2% *w*/*w*. 

### 2.3. Analysis of LTE Using Liquid Chromatography–Mass Spectrometry (LC–MS)

For qualitative analysis of LTE’s phytochemical composition, a sample solution with a concentration of 1 μg/mL was prepared by dissolving the initial mobile phase of LTE, which was then injected into an LC-MS system. LC-MS analysis was performed with a Dionex Ultimate 3000 HPLC system (Thermo Fisher Scientific, Waltham, MA, USA) coupled with a MaXis 4G Q-TOF mass spectrometer (Bruker Daltonics Inc., Germany) that was equipped with an electrospray ionization (ESI) interface and was operated in negative mode. Chromatographic separation was achieved using a Synergi Hydro-RP (150 mm × 2.0 mm, 4.0 µm, 80A; Phenomenex, Torrance, CA, USA) and water containing 0.1% acetic acid and methanol as the mobile phase, at a flow rate of 0.3 mL/min, with a gradient elution. The column temperature was maintained at 40 °C and the injection volume was 3 μL. The MS scan range was m/z 50–800 and the source parameters were set to: capillary voltage, −4500 V; end plate offset, −500 V; 2 bar nebulizer gas pressure; drying gas flow rate, 5 L/min; and dry temperature, 250 °C.

### 2.4. Animals and LTE Treatment

Only male mice were used in this study, as the estrous cycle is known to influence metabolism and exercise performance [30]. Twenty-week-old male C57BL/6N mice were purchased from Samtako (Osan, Korea) and housed in cages under standard conditions (22 ± 2 °C, 50–60% humidity, 12 h light–dark cycle) throughout the experiment. All mice were fed a standard laboratory chow diet ad libitum. LTE dissolved in phosphate-buffered saline (PBS) was administered once a day for 4 weeks to each test group via oral gavage at a dose of either 30 or 100 mg/kg. All animal experiments were performed in accordance with the Guide for the Care and Use of Laboratory Animals published by the US National Institutes of Health (NIH Publication No. 85–23, revised 2011). The protocol for the current study was approved by the Institutional Animal Care and Use Committee of Jeonbuk National University (Approval No. JBNU 2021-0125).

### 2.5. Moderate Intensity Treadmill Running Capacity 

Treadmill running was performed as described in our previous study with slight modifications [31]. Mice were acclimated to a single lane treadmill (Jeung Do Bio & Plant, Seoul, Korea) by performing a daily 30 min run at 10 m/min for 7 days prior to the exercise performance tests. Mice were then subjected to a daily moderate intensity treadmill running exercise test for 4 weeks. The moderate intensity running test began at a speed of 10 m/min for 10 min, followed by an increase in 2 m/min every 10 min to a maximum of 16 m/min until exhaustion. Exhaustion was defined as the inability to return to the treadmill running despite mild stimulation with a wooden cane. Running time and distance were recorded for each mouse during the last two weeks of the running test period.

### 2.6. Histology

Immediately after sacrifice, skeletal muscle tissues were placed in 30% sucrose solution and embedded with liquid nitrogen-cooled isopentane. For succinate dehydrogenase (SDH) staining, frozen sections of tissue (10 µm) were incubated in 0.2 M sodium phosphate-buffered solution (pH 7.6) containing 0.6 mM nitro blue tetrazolium and 50 mM sodium succinate (Sigma-Aldrich, St. Louis, MO, USA) for 30 min at 37 °C. Slides were washed with DiH_2_O and mounted with aqueous mounting media. For staining of myosin heavy chain isoforms, serial muscle sections were preincubated in a blocking solution of stock goat serum. The primary MyHC antibodies (MyHC I (#BA-D5), MyHC IIa (#SC-71) and MyHC IIb (#BF-F3), DSHB, Iowa City, IA, USA) were incubated overnight at 4 °C. After washing, secondary antibodies (Alexa Fluor 350-conjugated goat anti-mouse IgG2b (#A21140), Alexa Fluor 488-conjugated goat anti-mouse IgG1 (#A21121) and Alexa Fluor 594-conjugated goat anti-mouse IgM (#A21044), Thermo Fisher Scientific) were incubated for 1 h at 37 °C. For fiber-type specification, a region of the sections that contained approximately 200 fibers was selected. Images were acquired using a Leica DM750 microscope (Leica, Wetzlar, Germany). These fibers were then manually classified as immune-positive or immune-negative. We also calculated the cross-sectional area of the fibers using iSolution DT 36 software (Carl Zeiss, Oberkochen, Germany).

### 2.7. Indirect Calorimetry

Mice were housed in an Oxymax/CLAMS metabolic cage system from Columbus Instruments (Columbus, OH, USA) with one mouse/chamber. Mice were placed in metabolic cages for one day to adapt and avoid stress during analysis. After 24 h of acclimatization, mice were monitored continuously for 72 h with ad libitum feeding in an environmental room set at 20–23 °C with a 12–12 h (7:00 pm–7:00 am) dark–light cycle. The respiratory exchange ratio (VO_2_/VCO_2_) was measured using an Oxymax system. Data collected over the last 24 h of the experiment was used for analysis. 

### 2.8. Biochemical Analysis

Plasma levels of alanine aminotransferase (ALT) and aspartate aminotransferase (AST) were analyzed using appropriate kits (Biovision, Milpitas, CA, USA). 

### 2.9. Cell Culture 

C2C12 cells were obtained from the American Type Culture Collection (ATCC, Manassas, VA, USA). C2C12 myoblasts were maintained in a DMEM-supplemented culture with 10% FBS at less than approximately 80% confluence. Differentiation of C2C12 cells was initiated by replacing 10% FBS with 2% horse serum (Gibco Life Technologies, Waltham, MA, USA). The differentiation medium was changed every 2 days and cells at day 5 were considered as differentiated myotubes. LTE was exposed to cells for 5 days in the differentiation medium. 

### 2.10. Western Blotting

Cell or tissue homogenates (20 μg) were separated using 10% SDS-PAGE and transferred to PVDF membranes. After blocking with 5% skim milk, blots were probed with primary antibodies against CREB (#9197), *p*-CREB (#9198), ATF2, *p*-ATF2, P38, *p*-p38 (Cell Signaling Technology, Beverly, MA, USA), T-OxPhos (ab110413), Mfn1 (#ab567602) (Abcam, Cambridge, UK), OPA1 (#612606, BD Biosciences, Franklin Lakes, NJ, USA), HSP90 (#ADI-SPA-836-F, Enzo Life Sciences, Plymouth Meeting, PA, USA), PGC-1α (#AB-3242, Millipore, Danvers, MA, USA), α-myosin (#M4276) (Sigma-Aldrich), Drp1 (#sc-271583), Fis1 (#sc-376447), Nor1 (#sc-393902), Nur77 (#sc-365113), and Lamin B1 (#sc-6216, Santa Cruz Biochemicals, Dallas, TX, USA). HSP90 was used as a loading control. The membranes were briefly washed and then incubated with horseradish peroxidase-conjugated IgG (Zymed, South San Francisco, CA, USA) for 1 h at room temperature. Antibody signals were detected using a Las-4000 imager (GE Healthcare Life Science, Pittsburgh, PA, USA).

### 2.11. RNA Isolation and Real-Time Quantitative RT-PCR (qPCR)

Total RNA was extracted from skeletal muscle tissues using TRIzol reagent (Invitrogen, Carlsbad, CA, USA). First-strand cDNA was generated using the random hexamer primer provided in a first-strand cDNA synthesis kit (Applied Biosystems, Foster City, CA, USA). Specific primers for each gene (Appendix A) were designed using PrimerBank (https://pga.mgh.harvard.edu/primerbank). qPCR reactions were conducted in a final volume of 10 μL containing 10 ng of reverse-transcribed total RNA, 200 nM of forward and reverse primers, and PCR master mix. qPCR was performed in 384-well plates using an ABI Prism 7900HT Sequence Detection System (Applied Biosystems). The mRNA level of each target gene of interest was normalized to that of *Gapdh* (in the case of nuclear-encoded genes) or 16S rRNA (in the case of mtDNA-encoded genes).

For mitochondrial DNA content analysis, total DNA was extracted using a genomic DNA purification kit (Qiagen, Hiaden, Germany). Relative mtDNA was quantified by qPCR using primers for the mitochondrially encoded gene cytochrome oxidase 2 (*Cox2*), normalized to the nuclear-encoded gene cyclophilin A (*Ppia*).

### 2.12. Statistical Analysis

Data are expressed as the mean ± standard error of the mean (SEM). All data were tested for normality using the Shapiro–Wilk test and equal variance using Levene’s homogeneity test. For data that did not pass normality testing, log transformation was applied to generate a Gaussian-distributed dataset that could be subjected to a non-parametric Kruskal–Wallis test followed by Dunn’s comparison test. Statistical comparisons among multiple groups were made using one-way analysis of variance followed by Bonferroni’s post hoc analysis. The significance of differences between two groups was determined using Student’s unpaired *t*-test. A *p* value of less than 0.05 was considered significant. All analyses were performed using GraphPad Prism 9.4 software (San Diego, CA, USA).

## 3. Results

### 3.1. LTE Supplementation Enhances Endurance Exercise Performance

We analyzed LTE using LC-MS to identify its functional ingredients. The compounds were identified based on their mass spectra and by comparison with standards discussed in the previously published literature [29] (Figure 1A). Compounds **1**–**6** were identified as (**1**): epicatechin (EC), (**2**): (−)-epigallocatechin-3-gallate (EGCG), (**3**): myricetin-3-O-β-D-galactopyranoside, (**4**): myricetin-3-O-α-L-rhamnopyranoside, (**5**): myricetin-3-O-(2″-O-galloyl)-α-L-rhamnopyranoside, and (**6**): myricetin-3-O-(3″-O-galloyl)-α-L-rhamnopyranosid (Figure 1B).

To evaluate the effect of LTE on exercise performance, 20-week-old male C57BL/6N mice were subjected to daily exercise-to-exhaustion testing over 4 weeks, during which mice were administered either 30 or 100 mg/kg of LTE via oral gavage (Figure 2A). Both exercise training and LTE supplementation for 4 weeks did not affect body weight or food intake compared to the vehicle-treated group (Figure 2B,C). Furthermore, no differences were observed in liver damage (AST and ALT) or the tissue weight of liver, fat, and skeletal muscles between the LTE-supplemented and vehicle-treated groups (Appendix A). Mice treated with LTE 30 mg/kg and LTE 100 mg/kg demonstrated an increased running time by a significant 33 ± 6% and 80 ± 4%, respectively, and an improved running distance by 37 ± 8% and 94 ± 9%, respectively, compared to vehicle-treated mice (Figure 2D–F and Appendix A). However, supplementation with LTE did not cause meaningful changes to oxygen consumption (VO_2_), carbon dioxide production (VCO_2_), the respiratory exchange ratio (RER, VCO_2_/VO_2_), energy expenditure (EE), or heat production relative to the control mice (Appendix A).

### 3.2. LTE Supplementation Increases the Proportion of Oxidative Fibers

Enhanced endurance is generally associated with an increased proportion of oxidative myofibers (type I and IIa fibers) [9]. Thus, we examined the fiber-type composition of gastrocnemius (GAS) muscles by staining tissues with specific antibodies against MyHC isoforms. The result revealed that LTE supplementation significantly increased the number of type I myofibers in GAS muscles by 57 ± 2% and decreased the number of type IIb/IIx fibers by 14 ± 6% over vehicle-treated trained mice (Figure 3A). The size of cross-sectional gastrocnemius muscle fibers was largely reduced in the LTE-100 group compared to the vehicle control (Figure 3B). Specifically, the average size of type I myofibers was not affected by LTE treatment, whereas type II myofibers were reduced in size compared to control-group muscles (Figure 3C and Appendix A). We then compared the oxidative capacity of skeletal muscle by measuring SDH activity, also known as mitochondrial complex II (CII). Compared to vehicle control mice, GAS muscle fibers in LTE-treated mice showed a stronger staining reaction for SDH and the percentage of SDH-positive fibers was elevated in the LTE group by 46 ± 6% (Figure 3D). Similar results were found upon examination of the extensor digitorum longus muscles (Appendix A). Consistent with these changes, the GAS muscles of LTE-treated mice exhibited increased mRNA levels of type I fiber genes such as *Myh7* (496 ± 5%), *Tnni1* (564 ± 7%), *Tnnc1* (70 ± 5%), and *Tnnt1* (93 ± 6%), and decreased mRNA levels of type II genes such as *Myh4* (30 ± 6%) and *Tnni2* (25 ± 9%) (Figure 3E).

### 3.3. LTE Supplementation Increases Mitochondrial Content and Oxidative Capacity

It is well established that improved oxidative capacity of muscle fiber is associated with increased mitochondrial biogenesis and function [32]. Thus, we compared mitochondrial content and the expression of related genes in GAS muscles after LTE supplementation. Mitochondrial content, measured as the mitochondrial genome-to-nuclear genome ratio (mtDNA/nDNA), indicated that LTE treatment increased mitochondrial DNA (mtDNA) content by 132% (Figure 4A), which was confirmed by qPCR analysis of genes that are related to mitochondrial biogenesis, including *Mtco1*, *Mtco2*, and *Mcad* (Figure 4B). Western blotting analysis showed a significant increase in several key components of the mitochondrial electron transport chain complex, from CI to CV (ATP synthase), in LTE-treated GAS muscles (Figure 4C).

Because mitochondria are highly dynamic organelles and constantly alter their content and function through coordinated cycles of fusion, fission, and mitophagy [33], we measured the expression of genes associated with mitochondrial fusion–fission proteins in the GAS muscles. In contrast to the increases in the expression of mitochondrial biogenesis-related genes, LTE supplementation did not affect the protein levels of either mitochondrial fusion–fission proteins (such as OPA1, Mfn2, Drp1, and Fis1) or mitophagy-related proteins (LC3bII/I ratio) in the GAS muscles (Figure 4D), suggesting that LTE’s effect on exercise endurance is related to the regulation of mitochondrial biogenesis rather than mitochondrial dynamics.

### 3.4. LTE Supplementation Activates PKA–CREB–PGC1α Pathways

To delineate the molecular mechanisms which control fast-to-slow muscle fiber-type switching induced by LTE treatment, we analyzed the profiles of proteins involved in fiber-type specification. We first assessed the expression of key transcription factors and their regulators involved in mitochondrial biogenesis and maintenance of muscle integrity. Western blotting results indicated that the phosphorylation of CREB (49 ± 4%), ATF2 (160 ± 0%), p38 MAPK (258 ± 7%), Akt (49 ± 2%), and the protein and mRNA levels of PGC-1α (28 ± 1%), Mef2a (319 ± 9%), and Mef2c (131 ± 6%) were significantly increased in GAS muscles of LTE-treated mice compared to vehicle-treated mice (Figure 5A). We also found increased mRNA levels of genes downstream of CREB and of genes related to muscle fiber specification, such as *Nfatc1* (318 ± 4%), *Ppargc1a* (332 ± 7%), and *Nr4a3* (Nor1, 201 ± 8%) in the GAS muscles of LTE-treated mice (Figure 5B). Since PKA functions as an upstream kinase to activate CREB, we speculated that PKA may be activated by LTE. Western blotting for *p*-PKA substrates indicated that LTE increased PKA activity (Figure 5C), which was consistent with the increased level of *p*-CREB.

To identify the cell-intrinsic action of LTE on the fast-to-slow myofiber-type transition, we performed in vitro experiments. LTE did not affect the differentiation of C2C12 cells (Appendix A). Next, differentiating C2C12 cells were incubated with either 30 μg/mL of LTE or vehicle. Western blotting results for MyHC genes clearly indicated that LTE treatment induced fast-to-slow fiber-type transition and increased the number of type I and IIa fibers by 71 ± 4% and 51 ± 1%, respectively (Figure 6A). Furthermore, *Ppargc1a* (76 ± 8%) and *Nfatc1* (54 ± 6%) mRNA levels and OxPhos protein expression (25 ± 1% for CI, 92 ± 2% for CII, 47 ± 7% for CIII, 124 ± 6% for CIV, and 46 ± 7% for CV) were significantly increased by LTE treatment compared to vehicle (Figure 6B,C). To determine the causal link between LTE and activation of PKA and CREB-PGC-1α, we co-treated cells with LTE with or without PKA inhibitor H89 or Rp-cAMP and performed Western blot analysis. The previously observed LTE-induced increase in *p*-CREB and PGC-1α levels was completely nullified in the presence of H89 and partially suppressed by Rp-cAMP (Figure 6D and Appendix A), indicating that PKA activation led to phosphorylation and transactivation of CREB, and in turn, induction of PGC-1α.

## 4. Discussion

This study demonstrates that *L. tetragonum* increases exercise endurance by enhancing mitochondrial biogenesis in skeletal muscle. LTE supplementation was positively associated with molecular markers that indicate increased oxidative capacity of skeletal muscle. The observed improvements in exercise endurance were influenced by muscle remodeling, which is a feature of an exercise-trained phenotype [34]. With regard to muscle remodeling, LTE supplementation elevated the protein and mRNA levels of slow fiber-specific MyHC I, whereas fast fiber-specific genes MyHC IIb/IIx were either unchanged or downregulated in GAS muscles. Myofiber size analysis showed a significant reduction in the number of type II myofibers, but not type I myofibers. Given that oxidative fibers are smaller than glycolytic ones, this strengthens the argument that a shift in fiber-type occurred. SDH immunostaining results support these observations, indicating that LTE improves exercise endurance by increasing the proportion of slow muscle fibers relative to fast muscle fibers. Alterations in myofiber-type proportions have been reported in exercised humans as well. Gehlert et al., evaluated the proportions of different myofibers through muscle biopsies in young male cyclists who underwent cycling training for 3 months [35]. The proportion of type I myofibers tended to increase while IIa fibers decreased significantly post-training, although total training time did not correlate to the degree of fiber-type transition. In particular, people with a lower baseline of type I myofibers showed marked increases in type I fibers, supporting the importance of myofiber composition in human. LTE supplementation also increased mitochondrial content and function. These results jointly support the notion that LTE improved mitochondria biosynthesis and oxidative fiber formation, thereby increasing exercise endurance, as slow muscle fibers are rich in mitochondria and have a high oxidative metabolism. 

An important aspect of our study is that we used *L. tetragonum* which was produced in a smart-farming system with LED lamps in a highly controlled environment. Cultivation of *L. tetragonum* in the LED plant factory afforded several advantages, such as standardization of the plant extract (which guarantees reproducible bioactivity) and minimization of the risk of heavy metal contamination that is observed in *L. tetragonum* plants that grow along the coast of South Korea.

We identified the bioactive substances in the *L. tetragonum* extract, which were: (**1**) epicatechin (EC), (**2**) (-)-epigallocatechin-3-gallate (EGCG), and (**3**–**6**) myricetin glycosides (Figure 1A,B). Recently, Lee et al., identified more than 10 bioactive compounds in *L. tetragonum* ethyl acetate soluble fraction extract that produce hepatoprotective effects, including five of the substances described above, with the exception of EC [29]. A number of studies have analyzed the beneficial effect of EC on muscle health [36], and EC has been shown to enhance exercise endurance and increase both fatigue resistance and oxidative capacity in mice [37]. A recent clinical study also reported that EC may act as an exercise mimetic. In that study, ambulatory adult patients with Becker muscular dystrophy, a neuromuscular genetic disorder characterized by progressive loss of contractile skeletal muscle, were administered EC for a short period of time, which elevated mitochondrial content in muscles, increased the expression of PGC-1α and biomarkers indicative of mitochondrial biogenesis, and improved exercise endurance [38].

EGCG, a polyphenol compound found in green tea, has been extensively studied for its beneficial effect on muscle performance in animals and humans [39]. In male cyclists, EGCG consumption increased fat oxidation and exercise performance [39]. A previous study found that mice which were administered green tea extract containing EGCG for 10 weeks exhibited increased exercise endurance, improved energy metabolism, and enhanced fat oxidation [40]. The anti-fatigue effect of EGCG has also been demonstrated in mice subjected to swimming exercise [41]. In contrast, an in vitro study concluded that EGCG reduced slow-twitch muscle fiber generation and mitochondrial biosynthesis in C2C12 cells by inhibiting the expression of PGC-1α [42], leading to uncertainty regarding the benefits of EGCG. 

The benefit of myricetin has only recently been reported [43,44]. Myricetin was found to improve hypoxia-impaired physical performance in rats by maintaining mitochondrial biogenesis and increasing PGC-1α expression, but only in relatively high doses (50, 75, and 100 mg/kg per day for seven days) [44]. Myricetin has also been shown to promote fast-to-slow muscle fiber-type switching and improve exercise endurance capacity in rats [43], which can be attributed to the downregulation of transcriptional repressors of slow-twitch myofiber gene Sox6 [45]. The positive effect of LTE supplementation on exercise endurance capacity may be due to the collective influence of all six compounds that we identified in LTE. 

Mitochondrial mass can be evaluated using mtDNA content as well as expression levels of OxPhos and genes related to mitochondrial biogenesis. We found that mitochondrial mass increased after LTE supplementation, indicating that long-term LTE supplementation in tandem with treadmill exercise improves both mitochondrial biogenesis and oxidative capacity, leading to fast-to-slow fiber-type transition. In addition to mitochondrial biogenesis, mitophagy and mitochondrial dynamics contribute to mitochondrial homeostasis within muscle cells [46]. Any defect in these steps will lead to mitochondrial dysfunction and an imbalance of the types of myofibers. In this study, we found that LTE supplementation did not affect the expression of mitophagy and mitochondrial dynamics-related genes, which revealed that the positive effect of LTE on mitochondrial oxidative capacity is directly related to mitochondrial biogenesis rather than regulation of mitochondrial dynamics and mitophagy.

We observed that LTE markedly enhanced CREB phosphorylation in vivo and in vitro, and accordingly, increased the mRNA and protein levels of PGC-1α, a target of CREB. Signaling molecules that govern muscle plasticity, in particular, mitochondrial biogenesis, have been extensively studied over the past two decades, and the CREB/PGC-1α axis has been identified as a prominent pathway. PGC-1α is a master transcriptional regulator in mitochondrial biogenesis, as confirmed by the finding that PGC-1α expression is higher in slow muscle fibers than in fast muscle fibers [47]. Furthermore, PGC-1α transgenic mice exhibit increased mitochondrial content and oxidative capacity in skeletal muscle [48] compared to mice lacking PGC-1α, which demonstrate a less efficient oxidative muscle metabolism and reduced exercise endurance [49]. CREB is also a key factor in the metabolic function of skeletal muscle [50] and is reliant on transcriptional activation of PGC-1α. We found that slow-fiber activating molecules downstream of CREB, including PGC-1α and Nr4a3 (encoding Nor1), were markedly upregulated by LTE, evidencing the critical role of CREB in LTE’s beneficial effects. In addition, the fact that PKA inhibitor H89 completely prevented LTE-induced increases in CREB phosphorylation and PGC-1α expression in C2C12 cells, indicates that LTE activates PKA and intensifies CREB/PGC-1α axis activity. It remains unclear precisely how LTE activates PKA signaling. Intracellular cAMP levels are fine-tuned by the balance between cAMP producing adenylate cyclase and cAMP hydrolyzing phosphodiesterase. While the dependence of cAMP in PKA signaling is well established, the impact of LTE on cAMP homeostasis, and by extension, PKA–CREB signaling, warrants further investigation.

PGC-1α is regulated by transcriptional factors, and at a posttranslational level, by phosphorylation, acetylation, ubiquitination, and methylation [51]. Several signaling pathways have been identified in PGC-1α regulation, and we examined upstream signals that may be responsible for the activation of PGC-1α. Phosphorylation of PGC-1α can either activate or suppress its coactivator function. For example, p38 MAPK phosphorylates PGC-1α on Thr262, Ser265 and Thr298, which results in increased protein stability [52]. In contrast, Akt-mediated phosphorylation of PGC-1α inhibits its activity [53]. In this study, using phospho-specific antibodies, we observed that LTE supplementation increased the phosphorylation of both p38 MAPK and Akt in GAS muscles, suggesting that LTE supplementation may promote mitochondrial biogenesis partly via activation of the p38 MAPK/PGC-1α pathway.

Glucose and lipid are primary energy sources for skeletal muscles. After being subjected to daily moderate intensity exercise, our body energy shifts from preferring glucose to fatty acid as a primary energy source. This event ensures a constant supply of glucose to glucose-requiring organs and induces the body to burn off fat. Theoretically, if fatty acids were the only energy substrate being used, the RER would be equal to 0.7. Notably, however, we observed that RER values were not changed by moderate intensity exercise in conjunction with LTE supplementation. Environmental stimuli around the mice might explain these mismatched results.

## 5. Conclusions

Our results indicate that LTE supplementation improves exercise endurance of mice via increasing mitochondrial biogenesis and fast-to-slow myofiber-type switching in skeletal muscle. We observed that activation of PGC-1α downstream of PKA–CREB signaling increases mitochondria biosynthesis and promotes slow-twitch myofiber formation. In addition, we did not observe any adverse effects of LTE during the experimental period. However, we did not include a group of sedentary mice in this study and can offer only a limited interpretation of LTE’s effects on endurance exercise. In terms of human application, due to the presence of many environmental factors such as diet, alcohol, or drug use, it is generally difficult to expect the isolating effect of any dietary supplement on the exercise performance in humans. Future studies are warranted to validate the exercise-enhancing effect of LTE in humans.

## Figures and Tables

**Figure 1 nutrients-14-03904-f001:**
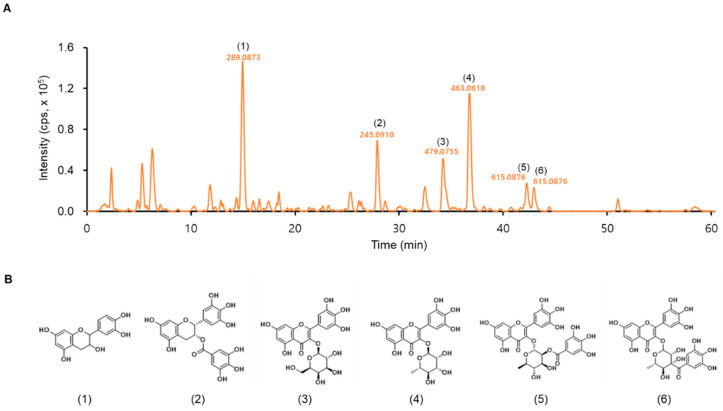
Analysis of *Limonium tetragonum* water extract (LTE). (**A**) LC/Q-TOF MS chromatogram (intensity measured in counts per second, cps) of LTE using negative ion mode. (**B**) Structures of the predominant substances in LTE. (**1**): epicatechin (**2**): (−)-epigallocatechin-3-gallate, (**3**): myricetin-3-O-β-D-galactopyranoside, (**4**): myricetin-3-O-α-L-rhamnopyranoside, (**5**): myricetin-3-O-(2″-O-galloyl)-α-L-rhamnopyranoside, and (**6**): myricetin-3-O-(3″-O-galloyl)-α-L-rhamnopyranoside.

**Figure 2 nutrients-14-03904-f002:**
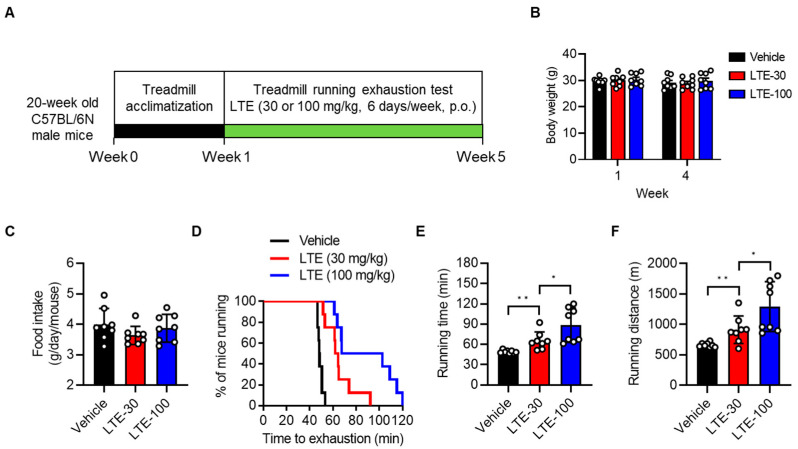
Effects of LTE supplementation on treadmill running endurance. (**A**) Schematic of experimental design to assess the effect of LTE on exercise performance. (**B**,**C**) Body weight change after the 4-week exercise and LTE supplementation program, and daily food intake of mice (*n* = 8). (**D**) Running population plotted against time to exhaustion (10 min at 10 m/min with an increase in running speed by 2 m/min every 10 min to a maximum of 16 m/min). Total running time until exhaustion was determined (*n* = 8). (**E**,**F**) Average running time and running distance until exhaustion under a forced running exercise-to-exhaustion test using data acquired during the last two weeks of testing (*n* = 8). Values are mean ± S.E.M., * *p* < 0.05 and ** *p* < 0.01. LTE-30, LTE 30 mg/kg p.o.; LTE-100, LTE 100 mg/kg p.o.

**Figure 3 nutrients-14-03904-f003:**
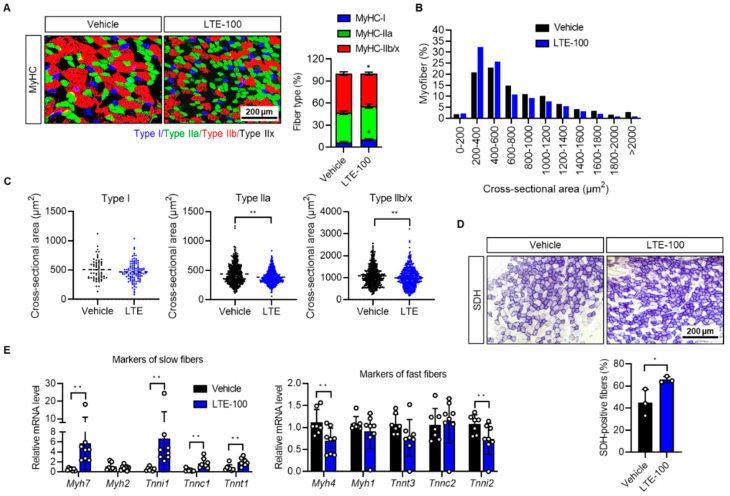
Effects of LTE supplementation on the proportion of muscle fiber-types. (**A**) Representative microscopic images of GAS muscles immunostained with anti-MyHC antibodies (types I, IIa, IIb, and IIx). The number of the different types of muscle fibers was counted manually (*n* = 5). Results are graphed as a percentage of the total number of fibers per muscle. (**B**) The quantification of the cross-sectional area of each type of myofibers in GAS muscles of mice is described in (**A**). (**C**) The cross-sectional area of each type of muscle fiber was determined based on the expression of MyHC-positive myofibers. (**D**) Succinate dehydrogenase (SDH) activity staining was performed on sections of GAS muscle (*n* = 3). (**E**) mRNA levels of known markers of type I and type II fibers were analyzed by qPCR. mRNA levels were standardized against *Gapdh* and plotted relative to the expression in vehicle-treated mice (*n* = 7–8). Values are mean ± S.E.M., * *p* < 0.05 and ** *p* < 0.01. LTE-100, LTE 100 mg/kg p.o.

**Figure 4 nutrients-14-03904-f004:**
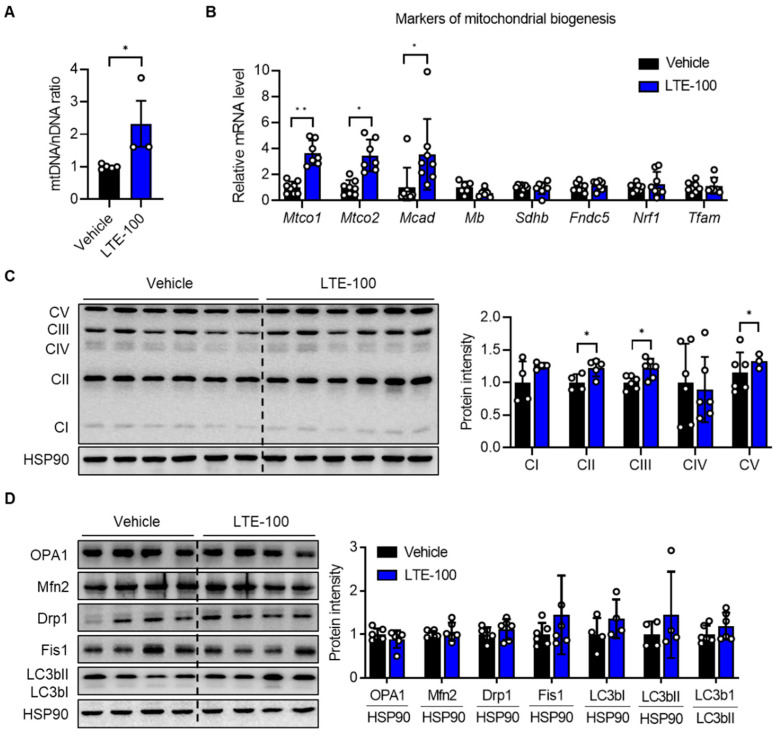
Effects of LTE supplementation on mitochondrial biogenesis**.** (**A**) Mitochondrial DNA (mtDNA) was quantified by qPCR using nuclear DNA (nDNA) as a standard (*n* = 3–5). (**B**) qPCR analysis of genes related to mitochondrial biogenesis in GAS muscles (*n* = 6–8). The expression of each gene was normalized with housekeeping gene *Gapdh*, whereas the expression of mitochondrial genome-encoded genes *Mtco1* and *Mtco2* was normalized with *Ppia*. (**C**) Western blotting analysis of the expression of OxPhos subunits and quantification of the intensity of OxPhos subunits relative to vehicle. (**D**) Western blotting analysis of mitochondrial fission and fusion genes. Western blot values are mean ± S.E.M (*n* = 3–6). * *p* < 0.05 and ** *p* < 0.01. LTE-100, LTE 100 mg/kg p.o.

**Figure 5 nutrients-14-03904-f005:**
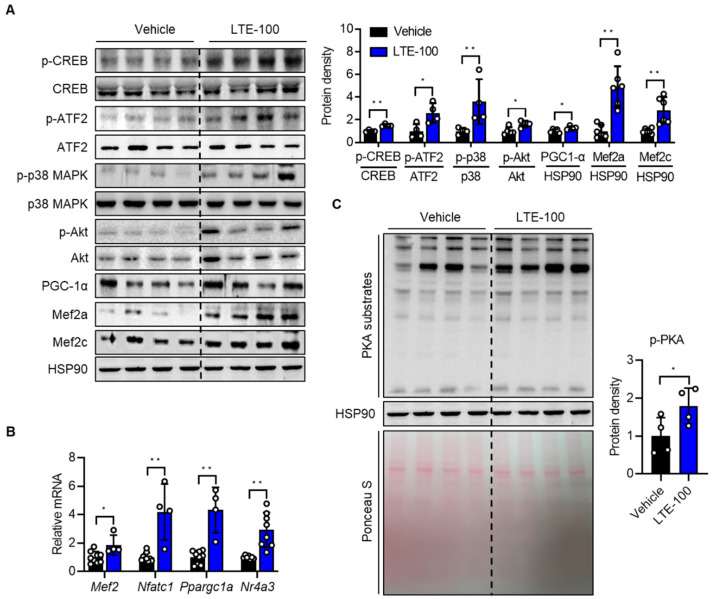
Effects of LTE supplementation on the signaling pathways leading to mitochondrial biogenesis and function**.** (**A**) Western blotting analysis of the genes involved in mitochondrial biogenesis in GAS muscles. Quantification results are shown (*n* = 4–6). (**B**) qPCR analysis of genes related to mitochondrial biogenesis (*n* = 4–9). Expression of each gene was normalized with housekeeping gene *Gapdh*. (**C**) Western blotting of PKA substrates. The red Ponceau S-stained gel is shown as loading control. Values are mean ± S.E.M. * *p* < 0.05 and ** *p* < 0.01. LTE-100, LTE 100 mg/kg p.o.

**Figure 6 nutrients-14-03904-f006:**
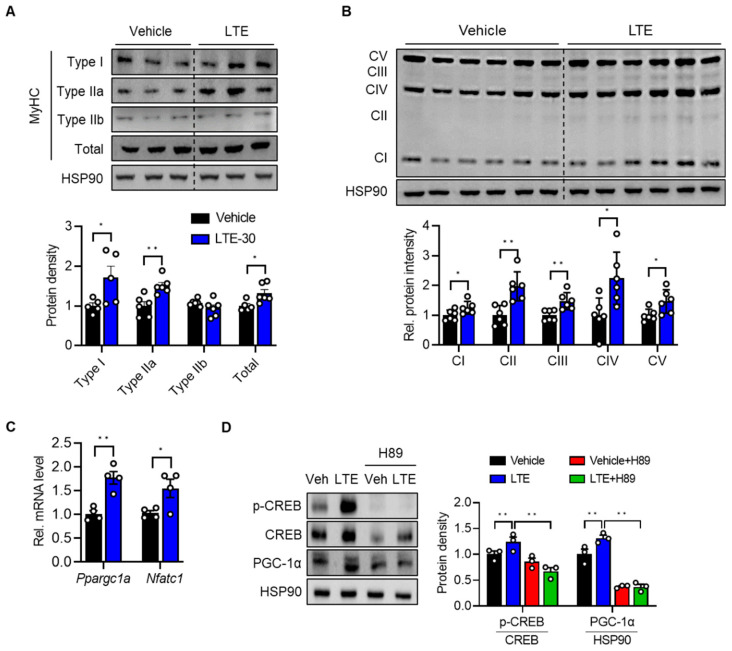
Effects of LTE treatment of C2C12 cells. C2C12 cells were incubated with LTE 30 μg/mL for 5 days in the differentiation medium. (**A**) Western blotting analysis of MyHC isoforms. Quantification results are shown (*n* = 4–6). (**B**) Western blotting for OxPhos proteins (*n* = 6). (**C**) qPCR analysis of the expression of *Ppargc1a* (encoding PGC-1α) and *Nfatc1* (*n* = 4). (**D**) Western blotting analysis after co-treatment of H89 (10 μM) and LTE. LTE, LTE 30 μg/mL; MyHC, myosin heavy chain; PGC-1α, peroxisome proliferator-activated receptor γ coactivator 1α; *Nfatc1*, nuclear factor of activated T cell 1. Values are mean ± S.E.M. * *p* < 0.05 and ** *p* < 0.01.

## Data Availability

The datasets generated during and/or analyzed during the current study are available from the corresponding authors upon reasonable request.

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
