# Peer review of "Limonium tetragonum Promotes Running Endurance in Mice through Mitochondrial Biogenesis and Oxidative Fiber Formation"

_nutrients, 2022, doi:10.3390/nu14193904_

Round 1
Reviewer 1 Report
Autohrs submitted a paper entitled: Limonium tetragonum, cultivated in a smart-farming system, promotes exercise performance in mice through mitochondrial biogenesis and oxidative fiber formation. The paper is well written and organized, it is appreciable sentence structures for clarity and conciseness, which is key to readability.
I find it of interest for anyone approaching this specific field of research and importantly, for the simple language used and the multiplicity of the pathways explored, I find the paper also suitable for disseminating into researchers.
However some points need to be considered.
11) The Abstract points out that LTE has potential clinical application as exercise mimetics but no direct evidence of LTE as exercise mimetic is provided. I would suggest to tone down the sentence.
22) The background in the introduction section is not supported by a comprehensive bibliography. For istance:
- - Line 36 is missiong of a proper reference;
- -Ref [4] is not completely exaustive for sentence in lines 40-41.
33) Please state why the animal testing was conducted only in male mice.
44) In the Results section, daily exercise to exhaustion testing over 4 weeks is described. Please define exhaustion in the appropriate Methods section. Moreover, which time point is plotted in Fig. 2D?
55) In line 198 it is reported that “tissue weight of liver, fat and skeletal muscles” is unchanged in LTE compared to vehicle group. How the authors explain the animals weight loss after 4 weeks exercise? Please clarify this point or provide data.
66) Missing label of panel A in Fig. 5.
77) Legend of Fig. 6 refers to panel E, but there is no panel E in the Fig. 6.
Author Response
1) The Abstract points out that LTE has potential clinical application as exercise mimetics but no direct evidence of LTE as exercise mimetic is provided. I would suggest to tone down the sentence.
Response: In response to reviewer’s comments, we tone downed the last sentence.
2) The background in the introduction section is not supported by a comprehensive bibliography. For instance:
- Line 36 is missing of a proper reference
-Ref [4] is not completely exhaustive for sentence in lines 40-41.
Response: In response to reviewer’s comments, we cited the reference articles appropriately.
3) Please state why the animal testing was conducted only in male mice.
Response: We used male mice to exclude hormonal effects. Indeed, estrous cycle is known to influence exercise performance and metabolism of mice (Aguiar AS Jr et al., Sci Rep. 2018; 8:10742). We described this point in the “Materials and Methods” section.
4) In the Results section, daily exercise to exhaustion testing over 4 weeks is described. Please define exhaustion in the appropriate Methods section. Moreover, which time point is plotted in Fig. 2D?
Response: We defined the exhaustion in the “Materials and Methods” section. In Figure 2D, we recorded the time when a mouse became exhausted. Y-axis presents the percent of mice with maximal running time until exhaustion.
5) In line 198 it is reported that “tissue weight of liver, fat and skeletal muscles” is unchanged in LTE compared to vehicle group. How the authors explain the animals weight loss after 4 weeks exercise? Please clarify this point or provide data.
Response: After the 4-week exercise, there were body weight loss in all three groups with no statistical significance (Vehicle, -0.98±1.04; LTE 30 mg/kg, -1.23±0.92; LTE 100 mg/kg, -0.48±1.41). We only measured wet weights of eWAT, liver, GAS muscle, and TA muscle. Differences in other organs or other parts of fat and muscle tissues may cause a slight difference of body weight loss in LTE-100 groups.
6) Missing label of panel A in Fig. 5.
Response: Thank you for pointing out our mistake.
7) Legend of Fig. 6 refers to panel E, but there is no panel E in the Fig. 6.
Response: Figure 6E was moved to graphical abstract.
Reviewer 2 Report
This work studied an important issue for muscle biology and most of the approaches used to answer the main question seem to be adequate. However, Material and Methods section is not properly described, figures should be improved and reorganized. Text should be also corrected. Introduction and discussions sections are poor and information to attract readers and show the originality of the work is missing. One important issue is about the “Maximal running test period”. As mentioned on pg 3 – line 114, “Mice were then subjected to a daily maximal treadmill running exercise test for 4 weeks.” . This is characterized as a “daily, maximal exercise training.” The most critical issue is that title, introduction, conclusion and all sections of the paper should consider the synergistic effects of LTE and exercise training. The authors do not evaluate the isolate (single) effects of LTE. LTE cannot be considered as an exercise mimetic because it was administered during exercise training and not in sedentary condition. Curiously, authors mentioned in the end of INTRO that: “administered LTE to C57BL/6 mice which were then subjected to a chronic exercise training program,…”, suggesting that authors already know about that. Authors should clearly discuss the findings IN VIVO and IN VITRO – similarities and differences. The most important finding describing the signaling pathway mediating the effects of TLE was performed with ONLY one NON-specific inhibitor of PKA (see below) and further experiments are needed to confirm the “strong conclusion” about this kinase. This reviewer has many doubts (below) that could help to improve the quality of the work.
Title:
1) Title does not mention the synergistic effect of LTE and exercise training
Abstract:
1) Provide more detail about treadmill exercise. See question 1 in my comments about METHODS.
2) Provide % of alteration in analyzed parameters.
3) Provide info about IN VIVO and IN VITRO experiments. For example, IN VITRO experiment in cells were not mentioned.
4) Some important findings like PKA inhibition in cells were not presented.
Introduction:
1) Authors should present data about the effects of exercise training and how LTE could exacerbate them.
2) Intracellular signaling pathways were poorly described.
Methods:
1) “Maximal treadmill running capacity” is not clear. There is a big mistake. “Maximal running test period” is a a “daily, maximal exercise training.” Mice performed maximal bouts of exercise every day and this condition could induce adaptation. Authors should consider in the whole paper that mice performed an exercise training protocol and performance was analyzed.
2) It is not clear why authors chose 16 m/min as a maximum speed in treadmill. How was this intensity defined? In our hands, this running speed is an intensity close to 50% of maximum speed, which is light to moderate intensity. Please, justify the use of these speeds and protocol.
3) Please, provide more info about “staining of myosin 125 heavy chain isoforms”. What did you do before adding primary antibodies? Blocking? Which protocol was used for it? Provide a reference. Magnification? Number of fibers analyzed? Whole muscle? Manual or automatic analysis of muscle sections!?
4) Please, provide dilution of antibodies in Supplementary file
5) In “2.8. Western blotting”section, describe the “loading control” used. Consider at least 2 proteins as loading control.
6) Please, analyze at least 2 genes as housekeeper.
7) What was the normality test performed? Which kind of data did you analyze: parametric or non-parametric ones?
8) The protocol to determine AST and ALT enzymes was not provided. It’s not clear if they were determined in serum/plasma or liver tissue… If they were analyzed in SERUM/PLASMA, they are NOT marker of liver function… Please, correct the info - Pg 5 – line 198
9) Please, explain why you chose the concentration of 30 μg/ml of LTE in IN VITRO (cell) experiment. Did you perform a concentration-response curve?
Results:
1) Please, provide quantitative data in the text (% of change).
2) CHECK ALL BLOTS ïƒ I highly recommend to reduce the exposure of western blots and reanalyze the data. Most of them are saturated. For example, OxPhos on Figure 4C… But check all of them.
3) Sub-sections and data/figures are confusing. I suggest to add SDH staining in Figure 4, because it shows mitochondrial enzymes like CII. Figure 3 should show only sarcomeric/contractile-related proteins.
4) Figure 2: Please, provide a Figure containing data about time to exhaustion/running time for EVERY DAY (daily) for EVERY GROUP. I suppose that exercise capacity improved even in Vehicle group during “training”.
5) Figure 2: What is the difference between D and E? It seems that E used another protocol, but it is not clearly described in Methods section.
6) Figure 2E-F: legend does not provide a clear information about the calculus to obtain these results. Are these data the average of 2 last weeks!?
7) Pg 5 – line 195 - “exercise training for 4 weeks induced body weight loss compared to before the 195 exercise program,” … Statical symbols are missing in Figure 2B or there is a mistake in some analysis.
8) Pg 5 – line 198 – “liver function (AST and ALT)”- see comments on METHODS.
9) Data about other muscles are presented in Supplementary file but they are not clearly described in text.
10) Pg 6 – line 220 - “Specifically, the average size 220 of type I myofibers was not affected by LTE treatment, whereas type II myofibers were 221 reduced in size compared to control-group muscles (supplementary Fig. 2)” – these data should be shown in MAIN FIGURES! Oxidative fibers are smaller than glycolytic ones, strengthening the argument that fiber type shift. Discuss (briefly) that in DISCUSSION section.
11) Sentence on pg 7 – lines 229-231 does not make sense and Figure 3E does not provide any relevant info to this work.
12) Pg 7 – line 250 – “genes that are re-250 lated to mitochondrial biogenesis (Fig. 4B)” – Which genes!? Sdhb (SDH)?! Be more precise!
13) Pg 8 - line 260 – “suggesting that LTE’s effect on 260 exercise endurance primarily arises due to the regulation of mitochondrial biogenesis ra-261 ther than mitochondrial dynamics.”- this sentence is not supported by the results. Strong phrase
14) Pg 8 – line 258 – “mi-258 tochondrial fusion-fission genes (such as OPA1, Mfn2, Drp1, and Fis1) or mitophagy-re-259 lated genes (LC3bII/I ratio)” … Replace word GENES by PROTEINS.
15) Figure 3A: Image is wrong. Please, apply the same magnification for both images. LTE fibers are clearly smaller than Vehicle. Bring some results about CSA from Supplementary to MAIN FIGURE.
16) Figure 3D: Split this Figure in 2 graphs – one for slow markers and another one for fast markers…Scale for Myh7 and Tnni1 mRNAs become difficult to analyze anything in fast mRNA markers.
17) LC3-I and -II as well as their ratio (already done) should be quantified.
18) Pg 9 - Excepting for PGC-1a, it’s not clear for the reader why CREB, ATF2, etc., were evaluated. Please, explain how these proteins interact in the intracellular signaling pathway and how they regulate mitochondria content and function. For example, p38 activates CREB that transcriptionally regulates PGC1a and NOR1… and more. This kind of info can be added in INTRO section.
19) Pg 9 – All proteins related to oxidative phenotype were up-regulated by TLE (i.e., Mef2, CREB and ATF2). Nfact1 was also upregulated. Why did PKA/CREB signaling (including target genes) was chosen to be investigated? The reason for that was not clearly presented.
20) Pg 9 – line 283 – “Since PKA is dependent on the 283 cellular cAMP and functions as an upstream kinase to activate CREB, we speculated that 284 PKA may be activated by LTE”- 1) What is the reason to mention cAMP?! TLE can activate GPCRs or inhibit Phosphodiesterase enzymes? And 2) Akt and p38 can also activate CREB! Why did you rule out their role?
21) Fig 5C: the densitometric analysis is missing. PKA substrates should be normalized to PONCEAU staining or other that reveal all proteins in the whole membrane.
22) Data about Akt protein is not mentioned in the text.
23) Total MyHC and HSP90 in Fig 6A is saturated… Please, see ALLL BLOTS in the paper to check saturation.
24) Pg 9 – line 296 – “PGC-1_α _levels was completely nullified in the presence of H89, indicating that PKA acti-296 vation led to phosphorylation and transactivation of CREB”- I suggest to replace word INDICATING by SUGGESTING, considering that H89 is NOT a specific inhibitor of PKA as demonstrated by several studies (e.g., Limbutara, K., Kelleher, A., Yang, CR. et al. Phosphorylation Changes in Response to Kinase Inhibitor H89 in PKA-Null Cells. Sci Rep 9, 2814 (2019).) It has been demonstrated that H89 can affect activities of protein kinases other than PKA, and therefore responses to H89 should not be regarded as sufficient evidence for PKA involvement in a signaling process. H89 should be used in conjunction with other PKA inhibitors, such as Rp-cAMPS. For this reason, I suggest that authors perform a similar experiment with another more specific PKA inhibitor.
25) Fig. 6D: Evaluate the same proteins Fig 5A, especially Akt and p38 to rule out that H89 inhibits these kinases and block other pathways.
26) “LTE was exposed to cells for 5 days in the differenti-140 ation medium.”, which may have improved differentiation process. Thus, I suggest that authors evaluate the protein content or gene expression of myogenic marker (i.e., MyoD, myogenin, Myf5, etc.) and perform a differentiation assay.
Discussion:
1) Authors discuss about bioactive substances identified in TLE in the beginning of DISCUSSION. HOWEVER, they did not study these compounds and lines 335 to 366 have many speculative data that could be omitted or, PART OF THEM, inserted in another part of the DISCUSSION. Authors SHOULD focus on their own results before speculating about compounds that they did not investigated.
2) Many proteins are “introduced” only in this section. This does not make sense to reader that does not understand the reason to investigate those proteins and signaling pathways. Please, reorganize the text.
3) Akt, p38 and PKA can activate CREB but this information is not clearly presented. (This message should be presented in INTRO instead of here)
4) Pg 12 – line 402 - “In contrast, Akt-mediated phosphorylation of PGC-1α inhibits its activity.” – reference is missing.
5) Pg 11 – line 339 – “A number 339 of studies have analyzed the beneficial effect of EC on muscle health” – Provide SOME references.
6) Authors did not discuss HOW TLE could activate PKA signaling. Does TLE activate GPCR receptors, inhibit a PDE enzymes and/or regulate any other upstream activator of PKA? Please, discuss about that.
7) Authors should discuss about the possibility of SYNERGISTC action of TLE during exercise training. What is the possible mechanism? They did not investigate TLE in sedentary mice and this should be clearly stated. May TLE have improved RECOVEY post-exercise? Is it a possible mechanism of action? Could TLE cause same effects in SEDENTARY mice?
Conclusion:
1) Be more precise. Is PKA-CREB signaling or p38-ATF2 the mediator of TLE effects!? Akt and p38 were activated by TLE and they stimulate CREB. In addition, H89 can inhibit Akt. Thus, H89 experiment cannot prove that PKA is the mediator of TLE effects.
Author Response
Title:
1) Title does not mention the synergistic effect of LTE and exercise training
Response: To observe the synergistic effect of LTE and exercise training, we should have included LTE supplementation group without exercise training. Because we only observed the combinatory effect of LTE and exercise training, we can’t use the term “synergistic effect”. Instead we changed the title as “Limonium tetragonum promotes running endurance in mice through mitochondrial biogenesis and oxidative fiber formation”.
Abstract:
1) Provide more detail about treadmill exercise. See question 1 in my comments about METHODS.
Response: Because of word limit, we just used the term “submaximal treadmill exercise”.
2) Provide % of alteration in analyzed parameters.
Response: We included % of alteration in each parameter.
3) Provide info about IN VIVO and IN VITRO experiments. For example, IN VITRO experiment in cells were not mentioned.
Response: We described in vitro results in the Abstract.
4) Some important findings like PKA inhibition in cells were not presented.
Response: We described the results of PKA inhibition study in the Abstract.
Introduction:
1) Authors should present data about the effects of exercise training and how LTE could exacerbate them.
Response: We described the health beneficial effects of exercise training.
2) Intracellular signaling pathways were poorly described.
Response: We described more in detail the signaling pathways leading to mitochondrial biogenesis.
Methods:
1) “Maximal treadmill running capacity” is not clear. There is a big mistake. “Maximal running test period” is a a “daily, maximal exercise training.” Mice performed maximal bouts of exercise every day and this condition could induce adaptation. Authors should consider in the whole paper that mice performed an exercise training protocol and performance was analyzed.
Response: We thank the Reviewer for commenting this issue. As commented, our exercise protocol is not a maximal treadmill exercise program, but a submaximal (moderate intensity) exercise program. We have now corrected our mistake throughout the manuscript.
2) It is not clear why authors chose 16 m/min as a maximum speed in treadmill. How was this intensity defined? In our hands, this running speed is an intensity close to 50% of maximum speed, which is light to moderate intensity. Please, justify the use of these speeds and protocol.
Response: Please refer to the above response.
3) Please, provide more info about “staining of myosin heavy chain isoforms”. What did you do before adding primary antibodies? Blocking? Which protocol was used for it? Provide a reference. Magnification? Number of fibers analyzed? Whole muscle? Manual or automatic analysis of muscle sections!?
Response: As commented, we provided more information for histological analysis.
4) Please, provide dilution of antibodies in Supplementary file
Response: We provided dilution of antibodies in Supplementary Table 2.
5) In “2.8. Western blotting” section, describe the “loading control” used. Consider at least 2 proteins as loading control.
Response: We used HSP90 as a loading control. We described this point in the Methods section. We also used GAPDH as a loading control. Overall results were the same.
6) Please, analyze at least 2 genes as housekeeper.
Response: For qPCR analysis, we used Gapdh as a housekeeping gene. When Rps was used, overall results were the same.
7) What was the normality test performed? Which kind of data did you analyze: parametric or non-parametric ones?
Response: We described about Shapiro-Wilk normality test and non-parametric Kruskal–Wallis test in the Methods section.
8) The protocol to determine AST and ALT enzymes was not provided. It’s not clear if they were determined in serum/plasma or liver tissue… If they were analyzed in SERUM/PLASMA, they are NOT marker of liver function… Please, correct the info - Pg 5 – line 198
Response: We described the protocol of AST and ALT analysis.
9) Please, explain why you chose the concentration of 30 μg/ml of LTE in IN VITRO (cell) experiment. Did you perform a concentration-response curve?
Response: We tested 10, 30, 100 μg/ml of LTE for in vitro experiments. We found 30 μg/ml of LTE was better than 10 μg/ml of LTE, but no difference with 100 μg/ml of LTE.
Results:
1) Please, provide quantitative data in the text (% of change).
Response: As commented, we included quantitative data in the Results section.
2) CHECK ALL BLOTS à I highly recommend to reduce the exposure of western blots and reanalyze the data. Most of them are saturated. For example, OxPhos on Figure 4C… But check all of them.
Response: We repeated WB analysis for several times. We are pretty sure that WB data represent the average protein levels among group.
3) Sub-sections and data/figures are confusing. I suggest to add SDH staining in Figure 4, because it shows mitochondrial enzymes like CII. Figure 3 should show only sarcomeric/contractile-related proteins.
Response: SDH staining indicates OxPhos capacity, but it also represents oxidative fibers.
4) Figure 2: Please, provide a Figure containing data about time to exhaustion/running time for EVERY DAY (daily) for EVERY GROUP. I suppose that exercise capacity improved even in Vehicle group during “training”.
Response: As commented, we provided running time data for every week in Supplementary Figure 2.
5) Figure 2: What is the difference between D and E? It seems that E used another protocol, but it is not clearly described in Methods section.
Response: In Figure 2D, we recorded the time when a mouse became exhausted. Y-axis presents the percent of mice with maximal running time until exhaustion. In Figure 3E, we measured average running time of each group.
6) Figure 2E-F: legend does not provide a clear information about the calculus to obtain these results. Are these data the average of 2 last weeks!?
Response: Yes, it was two-week average running time. We clearly indicated this point in the Figure 2 legend.
7) Pg 5 – line 195 - “exercise training for 4 weeks induced body weight loss compared to before the exercise program,” … Statistical symbols are missing in Figure 2B or there is a mistake in some analysis.
Response: We apologize for our negligence in this regard. We misinterpreted the data of body weight change. Both exercise training and LTE treatment did not affect body weight of mice. We corrected our mistake.
8) Pg 5 – line 198 – “liver function (AST and ALT)”- see comments on METHODS.
Response: We corrected our mistake.
9) Data about other muscles are presented in Supplementary file but they are not clearly described in text.
Response: We further analyzed EDL and soleus muscles and provided data in Figure S5. We also described these results.
10) Pg 6 – line 220 - “Specifically, the average size of type I myofibers was not affected by LTE treatment, whereas type II myofibers were reduced in size compared to control-group muscles (supplementary Fig. 2)” – these data should be shown in MAIN FIGURES! Oxidative fibers are smaller than glycolytic ones, strengthening the argument that fiber type shift. Discuss (briefly) that in DISCUSSION section.
Response: As commented, we moved the cross-sectional area data of each type of myofibers to Figure 3C. We also discussed this point in the Discussion section.
11) Sentence on pg 7 – lines 229-231 does not make sense and Figure 3E does not provide any relevant info to this work.
Response: We deleted Fig. 3E.
12) Pg 7 – line 250 – “genes that are related to mitochondrial biogenesis (Fig. 4B)” – Which genes!? Sdhb (SDH)?! Be more precise!
Response: We clearly indicated gene names in the text.
13) Pg 8 - line 260 – “suggesting that LTE’s effect on exercise endurance primarily arises due to the regulation of mitochondrial biogenesis rather than mitochondrial dynamics.”- this sentence is not supported by the results. Strong phrase
Response: We tone downed the sentence.
14) Pg 8 – line 258 – “mitochondrial fusion-fission genes (such as OPA1, Mfn2, Drp1, and Fis1) or mitophagy-related genes (LC3bII/I ratio)” … Replace word GENES by PROTEINS.
Response: As commented, we replaced the words with proper ones.
15) Figure 3A: Image is wrong. Please, apply the same magnification for both images. LTE fibers are clearly smaller than Vehicle. Bring some results about CSA from Supplementary to MAIN FIGURE.
Response: We used same magnification for both images. As responded in comment 10, we moved the cross-sectional area data of each type of myofibers to Figure 3C
16) Figure 3D: Split this Figure in 2 graphs – one for slow markers and another one for fast markers…Scale for Myh7 and Tnni1 mRNAs become difficult to analyze anything in fast mRNA markers.
Response: As commented, we split this Figure into two parts.
17) LC3-I and -II as well as their ratio (already done) should be quantified.
Response: We provided quantitative data of LC3b-I and LC3b-II.
18) Pg 9 - Excepting for PGC-1a, it’s not clear for the reader why CREB, ATF2, etc., were evaluated. Please, explain how these proteins interact in the intracellular signaling pathway and how they regulate mitochondria content and function. For example, p38 activates CREB that transcriptionally regulates PGC1a and NOR1… and more. This kind of info can be added in INTRO section.
Response: As commented, we introduced a number of different signaling pathways leading to PGC-1α activation.
19) Pg 9 – All proteins related to oxidative phenotype were up-regulated by LTE (i.e., Mef2, CREB and ATF2). Nfact1 was also upregulated. Why did PKA/CREB signaling (including target genes) was chosen to be investigated? The reason for that was not clearly presented.
Response: Please see the response to comment 18.
20) Pg 9 – line 283 – “Since PKA is dependent on the cellular cAMP and functions as an upstream kinase to activate CREB, we speculated that PKA may be activated by LTE”- 1) What is the reason to mention cAMP? LTE can activate GPCRs or inhibit Phosphodiesterase enzymes? And 2) Akt and p38 can also activate CREB! Why did you rule out their role?
Response: For the first question relating cAMP, we rephrased the sentence. We couldn’t rule out the role of Akt and p38 MAPK at this moment because LTE did not increase the protein levels of p-p38 MAPK and p-Akt in C2C12 cells (data not shown).
21) Fig 5C: the densitometric analysis is missing. PKA substrates should be normalized to PONCEAU staining or other that reveal all proteins in the whole membrane.
Response: As commented, blotted nitrocellulose membrane was stained with Ponceau S and used for the normalization of PKA substrates.
22) Data about Akt protein is not mentioned in the text.
Response: We described the changes of Akt in the text.
23) Total MyHC and HSP90 in Fig 6A is saturated… Please, see ALLL BLOTS in the paper to check saturation.
Response: We exposed Western blots with shorter time.
24) Pg 9 – line 296 – “PGC-1_α _levels was completely nullified in the presence of H89, indicating that PKA activation led to phosphorylation and transactivation of CREB”- I suggest to replace word INDICATING by SUGGESTING, considering that H89 is NOT a specific inhibitor of PKA as demonstrated by several studies (e.g., Limbutara, K., Kelleher, A., Yang, CR. et al. Phosphorylation Changes in Response to Kinase Inhibitor H89 in PKA-Null Cells. Sci Rep 9, 2814 (2019).) It has been demonstrated that H89 can affect activities of protein kinases other than PKA, and therefore responses to H89 should not be regarded as sufficient evidence for PKA involvement in a signaling process. H89 should be used in conjunction with other PKA inhibitors, such as Rp-cAMPS. For this reason, I suggest that authors perform a similar experiment with another more specific PKA inhibitor.
Response: We repeated the experiments in the presence of Rp-cAMPS and provided data in Supplementary Figure 7.
25) Fig. 6D: Evaluate the same proteins Fig 5A, especially Akt and p38 to rule out that H89 inhibits these kinases and block other pathways.
Response: Unlike in vivo tissue analysis (Fig. 5A), when we treated C2C12 cells with LTE, we couldn’t observe changes in p-p38 MAPK, p-Akt, Mef2a, and Mef2c (data not shown).
26) “LTE was exposed to cells for 5 days in the differentiation medium.”, which may have improved differentiation process. Thus, I suggest that authors evaluate the protein content or gene expression of myogenic marker (i.e., MyoD, myogenin, Myf5, etc.) and perform a differentiation assay.
Response: We cultured C2C12 cells for 5 days in the presence of LTE and performed Western blotting for myogenic markers. Results showed that LTE did not affect myoblast differentiation (Supplementary Figure 6).
Discussion:
1) Authors discuss about bioactive substances identified in LTE in the beginning of DISCUSSION. HOWEVER, they did not study these compounds and lines 335 to 366 have many speculative data that could be omitted or, PART OF THEM, inserted in another part of the DISCUSSION. Authors SHOULD focus on their own results before speculating about compounds that they did not investigated.
Response: Since LTE contains several polyphenols, we presume we can discuss the potential effect of each polyphenols.
2) Many proteins are “introduced” only in this section. This does not make sense to reader that does not understand the reason to investigate those proteins and signaling pathways. Please, reorganize the text.
Response: We believe that we sufficiently described the background of this study in the Introduction section.
3) Akt, p38 and PKA can activate CREB but this information is not clearly presented. (This message should be presented in INTRO instead of here)
Response: We introduced signaling pathway leading to CREB activation in the Introduction section.
4) Pg 12 – line 402 - “In contrast, Akt-mediated phosphorylation of PGC-1α inhibits its activity.” – reference is missing.
Response: We have cited appropriate reference.
5) Pg 11 – line 339 – “A number of studies have analyzed the beneficial effect of EC on muscle health” – Provide SOME references.
Response: We have cited appropriate reference.
6) Authors did not discuss HOW LTE could activate PKA signaling. Does LTE activate GPCR receptors, inhibit a PDE enzymes and/or regulate any other upstream activator of PKA? Please, discuss about that.
Response: We did not know how LTE activate PKA signaling. As reviewer commented, LTE could activate GPCR receptors or inhibit PDE. We discussed this point in the Discussion section.
7) Authors should discuss about the possibility of SYNERGISTC action of LTE during exercise training. What is the possible mechanism? They did not investigate LTE in sedentary mice and this should be clearly stated. May LTE have improved RECOVEY post-exercise? Is it a possible mechanism of action? Could LTE cause same effects in SEDENTARY mice?
Response: As reviewer pointed out, we did not include sedentary mice group. That’s why we couldn’t conclude that LTE has a synergistic or additive effects to endurance exercise. We described this point in the text.
Conclusion:
1) Be more precise. Is PKA-CREB signaling or p38-ATF2 the mediator of LTE effects!? Akt and p38 were activated by LTE and they stimulate CREB. In addition, H89 can inhibit Akt. Thus, H89 experiment cannot prove that PKA is the mediator of LTE effects.
Response: As shown by the WB analysis (Fig. 6D), PKA inhibitors H89 and Rp-cAMPS inhibited LTE-mediated CREB phosphorylation. These results suggest that LTE induces PGC-1α through PKA-CREB pathway. For the p-p38 MAPK and p-Akt signaling pathways, as responded to comment 28, no changes in these proteins were observed when C2C12 cells were treated with LTE. This discrepancy is thought to be the result of in vitro experiments failing to recapitulate in vivo experimental conditions. We therefore concluded that LTE induces PGC-1α through PKA-CREB pathway. Reflecting the above results, we rephrased the conclusion.
Reviewer 3 Report
With interest I have been reading the above mentioned manuscript. The manuscript contains interesting information and is well written. A few major concerns I would like to express:
1) You only use running time and running disctance as exercise outcome parameter. This is an interesting outcome but doesn't explain differences in oxidative and/or anaerobic performance. Therefore, it is mandatory to have more in depth information regarding oxygen uptake and carbon dioxide production and derived parameters like RER/RQ, Ventilatory threshold 1 and 2, peak VO2, etc. This is also important to analyse the daily max exercise tests.
2) As I understand well you do daily max exercise tests. Can you elaborate in more detail what effect these tests have on long-term performance outcomes? It could cause a kind of over-reaching/overtraining that effects energy metabolic pathways.
3) You study GAS and TA muscles that are mixed type of muscles. It would have been more informative if you would also have studies Soleus and EDL muscle that are 90% oxidative and 90% anaerobic, respectively in nature. Information from these muscle would be much more informative than just SDH muscle fiber histology as outcome parameter.
Author Response
1) You only use running time and running distance as exercise outcome parameter. This is an interesting outcome but doesn't explain differences in oxidative and/or anaerobic performance. Therefore, it is mandatory to have more in depth information regarding oxygen uptake and carbon dioxide production and derived parameters like RER/RQ, Ventilatory threshold 1 and 2, peak VO2, etc. This is also important to analyse the daily max exercise tests.
Response: We included indirect colorimetry data in the Figure S2. In contrast to the changes in the myofiber composition, we couldn’t observe the differences in VO2, VCO2, RER, energy expenditure, and heat generation among groups. We can’t explain this unexpected results at this moment. Generally, measurement of indirect calorimetry is best performed with the mice at steady state. Environmental stimuli around the mice affecting CO2/bicarbonate buffer system or respiration of mice might result in these mismatch results.
2) As I understand well you do daily max exercise tests. Can you elaborate in more detail what effect these tests have on long-term performance outcomes? It could cause a kind of over-reaching/overtraining that effects energy metabolic pathways.
Response: First of all, as reviewer 3 pointed out, our exercise protocol is not a “maximal exercise test”, but a “submaximal exercise test”. We therefore corrected our mistake. Glucose and lipid represent primary energy sources of skeletal muscle. Upon daily, submaximal treadmill (moderate-intensity) exercise, our body’s energy preference shifts from glucose to fatty acid as a primary energy source. This event keeps supplying a constant glucose to brain and other glucose-requiring organs (or cells) and also inducing the body to burn off excess fat thus reducing body fat. Theoretically, if fatty acids were the only energy substrate being used, the RER value would be equal to 0.7. However, in this study, RER values were not changed with submaximal exercise with LTE supplementation. As responded in comment 2, we can’t explain this unexpected results. We described this point in the Discussion section.
3) You study GAS and TA muscles that are mixed type of muscles. It would have been more informative if you would also have studies Soleus and EDL muscle that are 90% oxidative and 90% anaerobic, respectively in nature. Information from these muscle would be much more informative than just SDH muscle fiber histology as outcome parameter.
Response: As commented, we further analyzed EDL and soleus muscle tissues and added the results in Supplementary Figure 5.
Round 2
Reviewer 3 Report
With interest I have been reading the revised manuscript and must compliment all authors for the intensive work. I'm satisfied with some of the points addressed by me but still I'm not completely satisfied with the exercise protocols and testing.
You have the mice running submaximal and changed that in the manuscript but you run them until exhaustion. This is a contradiction. If you run till exhaustion then it can must considered maximal and not sub-maximal.
I still miss the individual indirect calorimetry data from the exercise 'tests' !!!
Author Response
With interest I have been reading the revised manuscript and must compliment all authors for the intensive work. I'm satisfied with some of the points addressed by me but still I'm not completely satisfied with the exercise protocols and testing.
You have the mice running submaximal and changed that in the manuscript but you run them until exhaustion. This is a contradiction. If you run till exhaustion then it can must considered maximal and not sub-maximal.
Response: We thank the Reviewer for identifying this issue. We used the term “moderate intensity exercise” instead of submaximal exercise.
I still miss the individual indirect calorimetry data from the exercise 'tests' !!
Response: In our previous revision, we included indirect colorimetry data in Figure S3. In contrast to changes in the myofiber composition, we did not observe any differences in VO2, VCO2, RER, energy expenditure, and heat generation among groups. We are unable to explain this unexpected result. Measurement of indirect calorimetry is best performed with the mice at steady state, as environmental stimuli around the mice that affect the CO2/bicarbonate buffer system or their respiration might result in these mismatch results. We described this point in the Discussion section.